# Teosinte (*Zea mays* ssp *parviglumis*) growth and transcriptomic response to weed stress identifies similarities and differences between varieties and with modern maize varieties

S. A. Bruggeman[1☯], D. P. Horvath[2☯]*, A. Y. Fennell[3], J. L. Gonzalez-Hernandez[3], S. A. Clay[3]

1 Biology Department, St. Augustana University, Sioux Falls, SD, United States of America, 2 USDA-ARS-ETSARC, Sunflower and Plant Biology Research Unit, Fargo, ND, United States of America, 3 Department of Agronomy, Horticulture & Plant Science, South Dakota State University, Brookings, SD, United States of America

☯ These authors contributed equally to this work.
* David.horvath@usda.gov

**Data Availability Statement:** All raw sequence files are available from the NCBI SRA database

## Abstract

Transcriptomic responses of plants to weed presence gives insight on the physiological and molecular mechanisms involved in the stress response. This study evaluated transcriptomic and morphological responses of two teosinte (*Zea mays* ssp *parviglumis*) (an ancestor of domesticated maize) lines (Ames 21812 and Ames 21789) to weed presence and absence during two growing seasons. Responses were compared after 6 weeks of growth in Aurora, South Dakota, USA. Plant heights between treatments were similar in Ames 21812, whereas branch number decreased when weeds were present. Ames 21789 was 45% shorter in weedy vs weed-free plots, but branch numbers were similar between treatments. Season-long biomass was reduced in response to weed stress in both lines. Common down-regulated subnetworks in weed-stressed plants were related to light, photosynthesis, and carbon cycles. Several unique response networks (e.g. aging, response to chitin) and gene sets were present in each line. Comparing transcriptomic responses of maize (determined in an adjacent study) and teosinte lines indicated three common gene ontologies up-regulated when weed-stressed: jasmonic acid response/signaling, UDP-glucosyl and glucuronyltransferases, and quercetin glucosyltransferase (3-O and 7-O). Overall, morphologic and transcriptomic differences suggest a greater varietal (rather than a conserved) response to weed stress, and implies multiple responses are possible. These findings offer insights into opportunities to define and manipulate gene expression of several different pathways of modern maize varieties to improve performance under weedy conditions.

(accession numbers PRJNA543241 and PRJNA543185).

**Funding:** DPH CRIS# 3060-21220-029-00-D, https://www.ars.usda.gov/research/project/?accnNo=429921, The funders had no role in study design, data collection and analysis, decision to publish, or preparation of the manuscript. SB, SC, Identification of differential agronomic traits in early stage teosinte, flint, dent, and sugar (sweet) corn varieties in competition with weeds. Hansen, S.A. $4,000. May 2013-April 2014. South Dakota State University Center for Excellence in Drought Tolerance Research Excellence in Plant Stress Research Awards. SB, SC, Identification of differential agronomic traits in early stage teosinte, flint, dent, and sugar (sweet) corn varieties in competition with weeds. Hansen, S.A. $7,400. April 1, 2013-March 30, 2014. South Dakota State University Research/Scholarship Support Fund 2013.

**Competing interests:** The authors have declared that no competing interests exist.

## Introduction

Domestication events leading up to the existence of modern-day maize varieties have been studied extensively by researchers interested in crop breeding and the evolution of agronomy, as well as archaeologists interested in mankind's contributions to the domestication of crops [1–11]. Research has determined modern day maize diverged from its wild progenitor teosinte about 9000 years ago [12–14].

Early producers of maize were Central American peoples in transition from hunter-gathers to farmers/agrarians. Ancient producers may have made plant selections for different microclimates, as an Incan Empire "agronomic field station" (Moray, near Cusco, Peru) has terraces that span 22 different microclimates [15]. Researchers have found maize and teosinte crosses highly susceptible to *Ustilago maydis*, a fungal pathogen of maize whose fungal spore sacks are a delicacy in Mexican cuisine [16]. This susceptibility may have contributed to the domestication of maize. When discarded seeds (from fruits or vegetables that were not eaten) from nomad kitchens grew back the next year, this aided in selecting the most favorable fruits and products. Because this was done with a relatively small selection of the wild population, a genetic bottleneck occurred, which reduced genetic diversity [17]. Approximately 30% of the genetic diversity in the original teosinte genome came through the genetic bottleneck and is found in today's maize hybrids and varieties [18].

Several key loci were involved in maize domestication, taking it from native teosinte habitat to cultivated maize fields. Physical statures of modern-day maize and teosinte lines are dissimilar, although teosinte produces edible kernels that grow readily in its native South and Central American habitat. A teosinte plant has many branches and glumes (ears). A modification to the *teosinte branched1* loci is responsible for the single main stalk found in maize [17]. Kernel fruit cases of teosinte are extremely hard. A modification of the *teosinte glume architecture 1* loci eliminated the hard coverings, exposing "naked" kernels, which allows for easy consumption [19, 20].

Researchers have investigated potential teosinte traits for maize improvement such as starch content, seed weight, oil content, and kernel count [9, 21]. Differences among teosinte rhizosphere and microbiome compositions across multiple climates and mechanisms associated with teosinte and maize response to pest and pathogen attack and defense also are being investigated as sources for plant health and genetic improvement [22, 23].

In organic and conventional maize production systems, weed presence can decrease maize yields up to 100%. Billions of dollars and thousands of hours of labor are spent each year to control weeds and optimize yield in maize systems worldwide. Increasing maize's ability to maintain yield in weed presence through understanding how weed stress impacts genetic expression and selecting genotypes that maintain expression observed under weed-free conditions may be another method of dealing with weeds. Literature suggests that sweet maize (*Zea mays* L. convar. *saccharata* var. *rugosa*) and some modern dent variants (field maize, *Zea mays* L.*indentata*) have varying degrees of weed tolerance, or the ability to suppress weeds [24, 25]. Under weed stress, maize typically decreases root growth and photosynthetic capacity [26–28], grows shorter, and decreases yield. Investigating weed response differences found in teosinte lines will contribute to determining genetic mechanisms available for increasing or building upon pre-existing crop tolerance abilities in crops.

The objectives of this study were to evaluate and compare morphologic and transcriptomic responses to weed stress between two teosinte lines when grown under field conditions and to identify similarities and differences in the transcriptomic response. This study serves as a preliminary guide for further investigations into the genes and pathways regulating the response to weed pressure that are both common and unique to maize and its wild progenitor and

**Table 1. Seed quality values for teosinte lines evaluated in 2014 at the South Dakota Research Farm, Aurora, South Dakota.**

| Line | Per 100 g Dry Matter | | | | |
|------|------|-----|-------|-----|------|
|      | **Prot** | **Fat** | **Fiber** | **Ash** | **Carb** |
| **Ames 21789** | 30.7 | 5.0 | 0.92 | 2.16 | 51.6 |
| **Ames 21812** | 26.4 | 5.4 | 0.87 | 2.24 | 55.3 |

Seed data based on Flint-Garcia et al., 2009. Abbreviations: Prot = Protein, Carb = Carbohydrate.

provides information needed to identify targets for manipulating this response for future improvement of maize lines.

## Materials and methods

### Field methods

Teosinte lines were selected based upon seed availability and previous research studies [21]. Teosinte lines were originally from the Guerrero, Mexico state, approximately 1100 km apart, and had varying seed characteristics (Table 1). Lines were from different altitudes, with one line collected from a warm, dry climate (Ames 21789), and the second from a cool, wet climate (Ames 21812), and differed in protein, fat, and CHO content (Tables 1 & 2). Teosinte lines were grown during the 2013 and 2014 growing seasons at the South Dakota State University Aurora Research Farm. Soil series was a Brandt silty clay loam (fine-silty, mixed, super-active, frigid Calcic Hapldolls). Additional information regarding the experimental location is available [29]. Lines were planted May 24, 2013 and May 30, 2014 in an amended four replication split-plot design, with teosinte line being the main treatment, and weedy or weed-free being the sub-treatment (amended referring to having weedy plots next to each other as often as possible to minimize labor). Plots were fertilized at a rate of 140kg N/hectare with urea treated with urease inhibitor the 1st week of June. Because seed numbers were limited (only 100 seeds from one of the varieties were available for the two year study), individual treatment plots consisted of a 3.6-m$^2$ area, with 6 seeds (2013) or 4 seeds (2014) planted in a 3-m x 3-m plot at an equilateral distance from each other and from plot borders. A naturally abundant weed population was allowed to grow unchecked in weedy plots, whereas weed free plots were maintained weed free by hand hoeing and weeding approximately once every 7–10 days during the growing season, which did not allow weeds to establish beyond the seedling stage.

### Data collection

On July 15, 2013 and 2014, about 6 weeks after planting (about the eight-leaf vegetative growth stage), data were collected for the number of established plants, plant height, and number of branches at plant base, as well as weed density data and weed biomass. Teosinte heights were measured with meter sticks from the soil surface to the top arch of the tallest leaf. Weed densities were measured by counting the number of individual plants in a 0.1 m$^2$ area in two

**Table 2. Climate information for seed source locations and experimental site.**

| Location | Line | Altitude (m) | Avg. Rainfall (mm) | Avg Temperature (C) | Avg. Daylength (hours) |
|----------|------|--------------|--------------------|--------------------|------------------------|
| Guerrero Mexico | Ames 21789 | 3 | 480 | 27 | 12 |
| 2 km west of Teloloapan | Ames 21812 | 1860 | 1000 | 22 | 12 |
| Aurora, SD, USA | NA | 190 | 660 | 15 | 14–15 |

separate locations within the same plot. Weed biomass was measured in these two density locations by clipping the plants present at their base and drying to constant weight at 60˚C before weighing.

In September of each growing season, branch number and plant height were evaluated, and final per plant biomass values were collected in October by cutting each individual plant bunch approximately 2 cm from the soil surface, storing in a large paper bag, drying at 60˚C to constant weight, and weighing.

## Statistical analysis

An amended split plot design was used with teosinte line as the main factor and weed presence (+/-) as the subplot factor. A pairwise, one-tailed t-test was performed on weedy and weed-free parameter data on a per-line basis to determine differences/similarities between the weedy and weed-free treatments. Parameter correlations with final biomass were determined using a step-wise regression model in the MASS program in R, and models are presented only if significant (p-value<0.1).

## Samples and RNA sequencing

In 2014, samples for transcriptome sequence analysis were acquired July 15 between the hours of 11 am and 2 pm (90 minutes before and after the sun's zenith), prior to obtaining detailed morphologic and growth characteristics. Tissue samples of the last 10 cm of the most recently emerged leaf from 3–4 representative plants per plot were combined in a collection tube (Falcon Plastics, 15x4 mm snap cap tube), and frozen in liquid nitrogen immediately for each biological sample. Four biological samples were collected from each treatment. Samples were stored in an ultra-cold -80˚C freezer until RNA extractions.

RNA extractions were performed using a modified pine tree extraction method utilizing Trizol (Invitrogen, Waltham, MA, USA) and Qiagen (Qiagen, Venlo, Netherlands) products, as cited [27]. RNA was assayed for quality and quantity utilizing a Nanodrop machine (Barnstead/Thermolyne, Dubuque, IA, USA). RNA samples were stored at -80 C˚ until library creation.

cDNA libraries were created following the NEBNext Ultra Directional RNA Library Prep Kit for Illumina protocol (New England Biolabs, Ipswich, MA, USA). cDNA libraries were sequenced at the University of Illinois genome labs, using Illumina paired or single end reads. Sequencing data (100 base paired end reads) were generated and the raw sequence data is available for teosinte lines Ames 21812 and Ames 21789 in the Sequence Read Archive at the National Center for Biotechnology Information (accession numbers: PRJNA543241and PRJNA543185, respectively). Sequence data was analyzed using the CLC Bio software program for *de novo* assembly (CLC Bio-Qiagen, Aarhus, Denmark). Illumina paired end reads were imported as fasta files. Default settings were used, except:1) read names were discarded; 2) minimum distance was 70; 3) maximum distance was 252; 4) quality scores was set to NCBI/Sanger Illumina Pipeline; and 5) ambiguous nucleotides was set to 2. Sequences were trimmed for quality for left and right read pairs per sample in "batch" mode. Fragments under 50 bp in length were discarded, however, broken pairs were saved. *De novo* assembly was performed by combining all resultant paired and trimmed files in the *De Novo* Assembly application. Automatic word size and bubble size was default, guidance only reads used a.cds file from *Zea mays* cv B73 (AGPv3) which was exported as a fasta file and imported back into CLC Bio.

The guided *de novo* assembly (S1 File) was exported as a fasta file and imported back in as the "reference sequence" for use in the RNASeq Analysis sub-program for differential gene expression. Reads were mapped back to the assembly, mismatch cost was set to 2, insert cost

was 3, deletion cost was 3, length fraction was 0.5, similarity was set at 0.8, and contigs were updated. RNASeq analysis was run as a batch on the paired-trimmed fasta files for each sample, with "calculate RPKM values for genes without transcripts" checked. "Expression level" was checked, and RPKM values were selected to be used as values.

In the Transcriptomics Analysis app folder, "Set Up Experiment" was used to find differential expression. An individual experiment was used to set up one experiment per line, to ease downstream data manipulation. For each experiment, the eight RNASeq Analysis output files (fasta) for each line were selected as input files. A "multi-group" "unpaired" experiment with 2 groups using existing expression values was indicated. Groups were named 789W (Ames 21789 weedy), 789C (Ames 21789 weed-free), 812W (Ames 21812 weedy), and 812C (Ames 21812 weed free), and groups were assigned by clicking on the correct column as designated.

The Quality Control application was run on resulting data. Box Plots, Hierarchical clusters, and PCA (Principal Component Analysis) were performed to determine if samples grouped with their treatments and to eliminate any outliers. Statistical analysis was performed on the experiment data using the Empirical analysis application. Assembly contigs were run against the maize protein database via the blastX program, and matched contig information was transferred to the differential expression dataset for further analysis. A gene expression data set, which only consisted of genes with at least 3 of 4 samples per treatment having RPKM values $\geq 5$, was used for Pathway Studio 9.0 (Elsevier Pathway Studio) analysis for Gene Set Enrichment Analysis (GSEA) and Sub-Network Analysis (SNA) for each line. Ontological representation was defined as significant if it met a p-value of less than 0.05 following a Fisher's Exact test for over-representation. MAPMAN was used to create hierarchical and non-redundant gene ontologies through analysis of DEGs to visualize gene expression associated with known metabolic processes.

## Results

### Climate data

Monthly temperature means were similar between years in June 2013 and 2014, and similar to the 30-year normal average (Table 3). July and September 2013 were warmer than in 2014 (9% and 21%, respectively). Both years were slightly warmer than the 30-year normal in August. Accumulated Growing Degree Days (GDD) (calculated using a 10˚C base) differed between years in the time period from planting to first sample collection (47% more GDD in 2013), and the overall growing season (27% more GDD in 2013) (Table 3).

Plots were not irrigated during the growing season, and natural rainfall differed by year (Tables 3 & 4). 2014 was wetter than 2013 in 2 of the 3 main growing season months (June and August) (Table 3), although by harvest (October), total precipitation in 2013 and 2014 was almost equal (Table 4).

**Table 3. Monthly temperature and precipitation averages with 30-year normal over 2013 and 2014 growing seasons.**

|  | Monthly Mean Temp˚C | | | Monthly Total Precip (cm) | | |
|---|---|---|---|---|---|---|
|  | 2013 | 2014 | 30 yr normal (1990–2010) | 2013 | 2014 | 30 yr normal (1990–2010) |
| June | 19 | 19 | 19 | 13 | 18 | 11 |
| July | 22 | 20 | 21 | 8 | 6 | 8 |
| August | 21 | 21 | 20 | 4 | 7 | 8 |
| September | 18 | 15 | 15 | 4 | 5 | 8 |

Abbreviations: Temp = Temperature; Precip = Precipitation; yr = Year.

**Table 4. Accumulated growing degree days (GDD) (base 10°C) and precipitation amounts (cm) for each time frame after planting until sampling (Planting to V8), from sampling until harvest (V8 to Harvest), and season long (Total).**

| | Planting to V8 | | V8 to Harvest | | Total | |
|---|---|---|---|---|---|---|
| Year | GDD | Precip (cm) | GDD | Precip (cm) | GDD | Precip (cm) |
| 2013 | 646 | 27 | 819 | 13 | 1465 | 40 |
| 2014 | 440 | 25 | 716 | 14 | 1156 | 39 |

Abbreviations: GDD = Accumulated Growing Degree Days; Precip = Precipitation.

## Weed density and weed biomass

In both growing seasons, weed densities differed between the two lines, but weed biomass was similar. In 2013, weed densities differed between the two lines, but weed biomass was identical (Table 5). In 2014 (sequencing data shown for this year), weed densities averaged from 85 ($\pm$2) plants/meter$^2$ in Ames 21789 to 285 ($\pm$16) plants/meter$^2$ in the Ames 21812 weedy plots, but weed biomass was similar in weedy plots between teosinte lines (averaged from 820 ($\pm$385) to 900 ($\pm$426) gm/m$^{2)}$) (Table 5). Weed-free treatments had no weeds present for biomass or density counts.

In 2013, the majority of weed biomass was attributed to naturally occurring weed species, including (in order of prevalence):common lambsquarters (*Chenopodium album* L.), velvetleaf (*Abutilon theophrasti* Medik), wild buckwheat (*Polygonum convolvulus* L.), and grasses, including yellow and green foxtail (*Setaria pumila* (Poir.) Roem. & Schult., *S. viridis* (L,) Beauv., respectively) (Figs 1 and 2). In 2014, foxtails and velvetleaf were the dominant weed species.

## Teosinte growth parameters

Teosinte response to weed pressure differed between years, as demonstrated by plant height, branch number and harvest biomass (Table 5). Teosinte plant height in July averaged 51 to 80-cm, and weedy/weed-free differences ranged from a height increase in weed stressed plants of 16% to a decrease of 36%. Ames 21789 demonstrated a 20% increase in weed-stressed plant height in 2013, whereas in 2014 this line was 45% shorter in weed-stressed conditions. Ames 21812 was similar in height under weed-free and weed-stressed conditions in 2013, but weed-stressed plants were 8% taller than the weed-free plants in 2014 (Table 5). Teosinte plant

**Table 5. Teosinte parameter averages for 2013 and 2014 seasons.**

| | | July | | | | September | | October | |
|---|---|---|---|---|---|---|---|---|---|
| | | Weed Density | | Teosinte | | Teosinte | | Teosinte | |
| Teosinte Lines | | Weed Density | Weed Biomass | Plant Height | Branches | Plant Height | Branches | Biomass | Biomass per Branch |
| Line/Year | Trt | Plants/m$^2$ | gm/m$^2$ | cm | #/plant | cm | #/plant | gm/plant | gm/branch |
| Ames 21789 | W | 30(1)* | 266(98)* | 59(2)* | 6(3) | 128(14)* | 13(6)* | 393(200)* | 31(6) |
| 2013 | WF | 0 | 0 | 51(9) | 8(3) | 156(8) | 25(0) | 883(216) | 36(11) |
| Ames 21789 | W | 85(2)* | 900(426)* | 51(12)* | 4(1) | 143(10)* | 3(1)* | 57(11.7)* | 18(4) |
| 2014 | WF | 0 | 0 | 80(7) | 5(1) | 158(9) | 28(5) | 766(103) | 27(2) |
| Ames 21812 | W | 60(7)* | 266(63)* | 73(10) | 3(1) | 155(7)* | 4(2)* | 315(191)* | 78(28) |
| 2013 | WF | 0 | 0 | 80(6) | 5(3) | 196(26) | 10(3) | 810(317) | 79(15) |
| Ames 21812 | W | 285(16)* | 820(385)* | 70(8) | 2.2(1)* | 167(22) | 1.4(1)* | 69(28.6)* | 59(10) |
| 2014 | WF | 0 | 0 | 63(12) | 5(1) | 182(20) | 11(0) | 580(117) | 56(29) |

* indicates significance at p<0.05. Numbers in parentheses indicate 95% confidence interval. Abbreviations: Trt = Treatment.

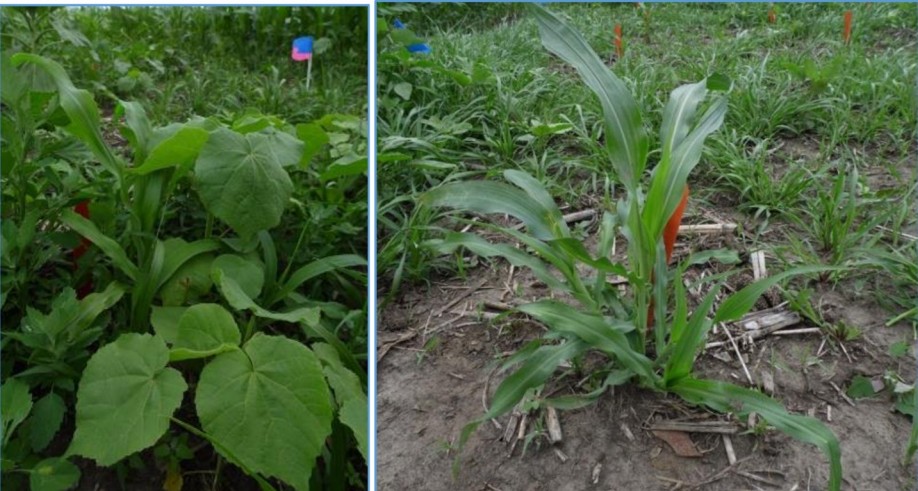

**Fig 1. Teosinte (*Zea mays ssp. parviglumis*) in weedy and weed-free environments, early season.** *Abutilon theophrasti* and *Setaria glauca* can be seen in the foreground (left) and background (right).

height in September was decreased by weed stress in both lines in 2013, but in 2014 only Ames 21789 had a 10% height reduction.

Branch number in July was unaffected both years in Ames 21789, and in 2013 in Ames 21812. In July 2014, Ames 21812 had a 56% decrease in weed-stressed plant branch number.

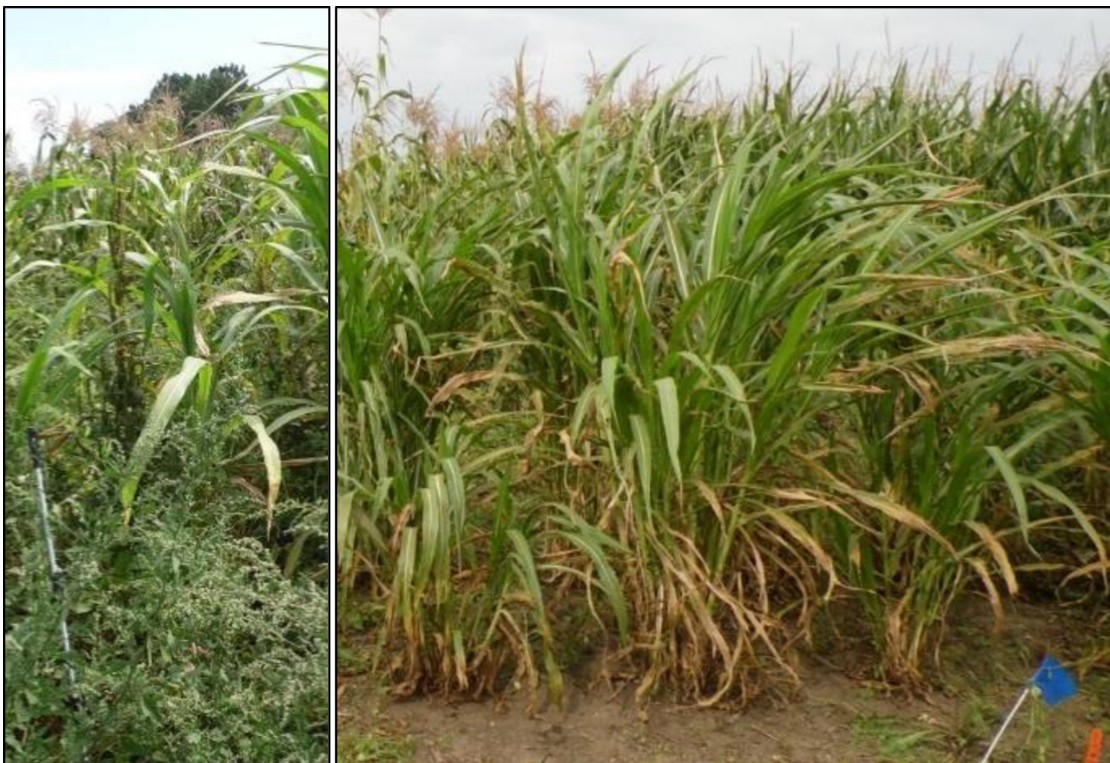

**Fig 2. Teosinte (*Zea mays ssp. parviglumis*) in weedy and weed-free environments, late season.** Common lambsquarters (*Chenopodium album*) can be seen in the foreground (left).

Planting density was low, and this allowed single plants to branch in an unrestricted manner, with some plants in the weed-free plots having diameters of 0.14 m$^2$. September branch number in both teosinte lines decreased in weed stressed plants in comparison to weed-free plants both years. Decreases in branch numbers for weed-stressed plants in contrast to weed-free plants ranged from 48–89%. In 2014, Ames 21812 averaged only 1 branch per plant in weed-stressed plants, compared to an average of 4 branches in 2013. In 2013, average branch loss was lower than in 2014 (54% and 85%, respectively).

Teosinte harvest biomass in weed-stressed plants was less than weed free plants due to decreased branch number and plant height. A 58% average biomass loss was measured in 2013 whereas in 2014 there was a 90% average biomass loss for both lines.

## Sequencing results and mapping of fragments

Twenty to forty-five million cDNA fragments were obtained for 16 different samples, resulting in 16 to 36 million paired end reads mapping to the assembly, with 13 to 27 million reads mapping uniquely (S1 Table). The guided assembly was comprised of 140,292 contigs of which 139,492 were 200 bp or longer, with 492 million reads.

## Differential gene expression

Ames 21789 had a total of 71 DEG's (Differentially Expressed Genes) between treatments, of which 61 were upregulated and 10 were downregulated in weed-stressed relative to weed-free treatments, with sequences of unknown function totaling 13. Ames 21812 had 32 DEG's with 22 upregulated and 10 downregulated genes in weed-stressed relative to weed-free treatments. Of these, 7 sequences were of unknown function. Only one DEG was shared between the two lines, GRMZM2G114751, a nodulin *MtN21* family protein, involved in transmembrane transport, which was up-regulated in both Ames 21789 and Ames 21812 weed-stressed plants in comparison to their weed-free counterparts (2.4 and 4.6-fold change, respectively) (S2 File). This gene was also differentially regulated in a similar maize study comparing weed-free to weed stressed maize, but demonstrated an opposite (up-regulated) expression pattern [28], and was not differentially expressed in the weed stressed treatments in maize in a similar greenhouse study [30].

## Gene functional category comparisons

Gramene/Ensemble ID's of DEG's were matched to corresponding MIPS (Munich Information Center for Protein Sequences) categories. Distribution among gene functional categories varied between the two teosinte lines (Table 6). The largest percentage of genes were classified in the "unknown" category, with 42% and 53% of total DEG's in Ames 21789 and Ames 21812, respectively. Hormone metabolism was the 2$^{nd}$ most common gene category in both lines, with 7% (Ames 21789) and 9% (Ames 21812) of total genes effected. The protein category in Ames 21789 also had 7% total genes affected but was not an affected MIPS category in Ames 21812. In Ames 21812, the RNA regulation category also had 9% of total genes affected, with no RNA-regulatory associated DEG's effected in Ames 21789.

Several other MIPS categories were affected in both Ames 21789 and Ames 21812, including UDP glucosyl and glucoronyl transferases, amino acid metabolism, copper and flavone oxidases, and unspecified development. Ames 21789 had between 2 and 4 DEG's effected for each category, whereas Ames 21812 only had 1 gene in each category mentioned.

MIPS categories unique to Ames 21789 which were not affected in Ames 21812 included protein secondary metabolism, glutathione-S transferases, redox (ascorbate and glutathione),

**Table 6. Distribution in MIPS categories of differentially expressed genes in Ames 21789 and Ames 21812 under weed stress in 2014.**

| MIP # | MIPS Category | Percent of total 21789 | Total Genes 21789 | Percent of total 21812 | Total Genes 21812 |
|---|---|---|---|---|---|
| 35 | unknown | 42.3 | 30 | 53.1 | 17 |
| 17 | hormone metabolism | 7.0 | 5 | 9.4 | 3 |
| 29 | protein | 7.0 | 5 | 0 | 0 |
| 16 | secondary metabolism | 5.6 | 4 | 0 | 0 |
| 26 | misc.UDP glucosyl and glucoronyl transferases | 5.6 | 4 | 3.1 | 1 |
| 26 | misc.glutathione S transferases | 5.6 | 4 | 0 | 0 |
| 21 | redox.ascorbate and glutathione | 4.2 | 3 | 0 | 0 |
| 34 | transport.amino acids | 4.2 | 3 | 0 | 0 |
| 13 | amino acid metabolism | 2.8 | 2 | 3.1 | 0 |
| 26 | misc.oxidases—copper, flavone etc. | 2.8 | 2 | 3.1 | 1 |
| 33 | development.unspecified | 2.8 | 2 | 3.1 | 1 |
| 1 | PS.calvincycle.GAP | 1.4 | 1 | 0 | 0 |
| 4 | glycolysis.unclear/dually targeted | 1.4 | 1 | 0 | 0 |
| 8 | TCA / org. transformation | 1.4 | 1 | 0 | 0 |
| 9 | mitochondrial electron transport / ATP synthesis | 1.4 | 1 | 0 | 0 |
| 11 | lipid metabolism.lipid degradation | 1.4 | 1 | 0 | 0 |
| 26 | misc.peroxidases | 1.4 | 1 | 0 | 0 |
| 30 | signalling | 1.4 | 1 | 3.1 | 1 |
| 20 | stress.biotic | 0.0 | 0 | 3.1 | 1 |
| 20 | stress.abiotic.heat | 0.0 | 0 | 3.1 | 1 |
| 3 | minor CHO metabolism | 0.0 | 0 | 3.1 | 1 |
| 27 | RNA.regulation of transcription | 0.0 | 0 | 9.4 | 3 |
| 28 | DNA.unspecified | 0.0 | 0 | 3.1 | 1 |

amino acid transport, Calvin Cycle, glycolysis, Tricarboxylic acid cycle, mitochondrial electron transport, ATP synthesis, lipid metabolism, and peroxidases.

Categories unique to Ames 21812 included biotic and abiotic stress, minor CHO metabolism, RNA regulation and unspecified DNA (Table 6).

## Gene Set Enrichment Analysis (GSEA) and Sub-Network Analysis (SNA)

Although the number of DEGs was small, a larger gene set could be analyzed via GSEA and SNA to provide a better indication of the physiological impact of weed interference in teosinte. A gene set consisting of 19,743 genes for both teosinte lines was analyzed with Pathway Studio for GSEA and SNA. Lists of overrepresented gene ontologies and sub-networks of upregulated genes in weed-stressed plants, down-regulated genes in weed-stressed plants, and of significantly (p-value < 0.05) affected genes were created (S3 File and Table 7).

In Ames 21789, there were 224 overrepresented gene set ontologies in up-regulated genes compared to weed-free plants. Of these 224 ontologies, 152 were shared with Ames 21812, which had 221 ontologies overrepresented (S3 File). Common ontologies upregulated in both teosinte weed stressed treatments included several regulatory proteins, a number of defense responses, and UDP glucosyl and glucuronyl transferases. Sixteen separate upregulated hormone related ontologies, including jasmonic acid, auxin, abscisic acid, salicylic acid, brassinosteroid, and ethylene ontologies were common to both teosinte lines.

While there were many common responses to weed stress among both teosinte lines, each line had unique enriched ontologies overrepresented in up-regulated gene sets (S3 File). Seventy-two ontologies were unique to up-regulated genes in Ames 21789, including a few

**Table 7. Subnetworks overexpressed through significant genes in weed-stressed teosinte plants compared with weed-free plants.**

| Subnetwork Description | Neighbors | Meas 789 | Meas 812 | Med Chg 789 | Med Chg 812 | Shared or Unique |
|---|---|---|---|---|---|---|
| PCRCP of defense response | 482 | 63 | 30 | 2.04 | 1.83 | S |
| PCRCP of lignification | 63 | 12 | 9 | 2.66 | 1.84 | S |
| PCRCP of plant defense | 323 | 41 | 23 | 2.19 | 1.89 | S |
| PCRCP of systemic acquired resistance | 87 | 15 | 10 | 2.12 | 1.60 | S |
| Expression Targets of COI1 | 55 | 11 | 0 | 4.57 | 0.00 | U789 |
| PCRCP of cell growth | 173 | 9 | 0 | 2.62 | 0.00 | U789 |
| PCRCP of jasmonate response | 63 | 15 | 0 | 2.62 | 0.00 | U789 |
| PCRCP of lignin biosynthesis trait | 51 | 8 | 0 | 2.26 | 0.00 | U789 |
| PCRCP of lignin content | 30 | 6 | 0 | 2.21 | 0.00 | U789 |
| PCRCP of lipid peroxidation | 38 | 8 | 0 | 2.19 | 0.00 | U789 |
| PCRCP of nodulation | 67 | 7 | 0 | 2.21 | 0.00 | U789 |
| PCRCP of response to osmotic stress | 74 | 8 | 0 | 2.67 | 0.00 | U789 |
| PCRCP of root length | 69 | 7 | 0 | 3.19 | 0.00 | U789 |
| PCRCP of ROS generation | 170 | 24 | 0 | 1.89 | 0.00 | U789 |
| PCRCP of somatic embryogenesis | 54 | 7 | 0 | 3.16 | 0.00 | U789 |
| PCRCP of cell death | 419 | 0 | 28 | 0.00 | 1.30 | U812 |
| PCRCP of disease resistance | 290 | 0 | 15 | 0.00 | 1.85 | U812 |
| PCRCP of ER unfolded protein response | 51 | 0 | 6 | 0.00 | 1.68 | U812 |
| PCRCP of hypersensitive response | 153 | 0 | 14 | 0.00 | 1.60 | U812 |
| PCRCP of membrane depolarization | 12 | 0 | 5 | 0.00 | 1.76 | U812 |
| PCRCP of plant immunity | 160 | 0 | 9 | 0.00 | 1.60 | U812 |
| PCRCP of response to ethylene stimulus | 127 | 0 | 11 | 0.00 | 1.85 | U812 |
| PCRCP of shoot growth | 86 | 0 | 11 | 0.00 | -1.19 | U812 |
| PCRCP of stomata development | 73 | 0 | 8 | 0.00 | 1.39 | U812 |
| PCRCP of transpiration | 65 | 0 | 8 | 0.00 | 1.76 | U812 |

Neighbors = number genes directly related to network, Meas is number of genes available for analysis in our dataset. p-value<0.05. 789 = Ames 21789, 812 = Ames 21812Meas = Measured, MedChg = Median Change, S = Shared between Ames 21789 & 21812, U789 = Unique to Ames 21789, U812 = Unique to Ames 21812. PCRCP = Proteins/Chemicals Regulating Cellular Processes.

common cellular responses-particularly those involved in nutrient stress and transport. Ames 21812 had 69 unique gene sets overrepresented among genes up-regulated during weed stressed, including a higher number of ontologies associated with oxidative stress than was observed in Ames 21789. Many unique responses were simply more gene sets or subnetworks affected in a certain category than in the other line (ie. jasmonic acid related, 2 more in Ames 21812 than in Ames 21789).

Down regulated ontologies in weed stressed plants in Ames 21789 totaled 148 (S3 File), and Ames 21812 had 140 downregulated, of which only 74 were shared between the two lines. Common ontologies down-regulated in weedy plants included numerous photosystem and photosynthesis related pathways and networks, several responses to light, and regulation of carbon fixation. Ames 21789 had 74 unique downregulated gene ontologies and Ames 21812 had 66 ontologies that were unique to that line. Ames 21789 appeared to have a greater number of ontologies associated with photosynthesis represented among the genes down-regulated by weeds than did Ames 21812. Likewise, Ames 21812 had more ontologies associated with growth and development than Ames 21789 among the genes down-regulated by weed interference.

When only significantly affected genes (q<0.05) were utilized in the GSEA analysis, the number of overrepresented ontologies was decreased. Ontologies involving significantly

affected genes included 102 in Ames 21789 (S3 File), and 79 in Ames 21812, of which 63 were common to both. Interestingly, every significantly affected gene ontology or subnetwork in both Ames 21812 and Ames 21789 were upregulated in weed-stressed plants, except 2: the sterol biosynthetic process and regulation of shoot growth (both were down-regulated) (S3 File).

Upregulated common significant gene ontologies included UDP gluco- and glycosyltransferase activity, quercitin 3/7-O-glucosyltransferase activity, and response to jasmonic acid, among others. Several defense responses, hormone related ontologies and subnetworks, membrane and transmembrane associated ontologies, numerous multi-purpose compounds, the MAPK cascade, and response to karrikin (a plant growth regulator found in smoke and associated with auxin signaling [31]) were all upregulated in weed-stressed teosinte lines. No significant ontologies for photosynthesis, chlorophyll, or photosystem was conserved between the two lines.

Subnetwork enrichment analysis was performed independently on the filtered gene set for both lines, yielding information regarding specific biological networks associated with weed stress response. There were 38 upregulated shared subnetworks between teosinte weed-stressed plants. Overall, Ames 21789 had 69 overrepresented subnetworks, of which 31 were unique (S3 File). Ames 21812 had 62 overrepresented upregulated subnetworks in weed stressed plants compared with weed-free plants, of which 24 were unique. Common up-regulated networks included binding partners of AGB1 (negatively regulates ABA response) and BAK1 (regulates brassinosteroid receptor BRI1), expression targets of COI1, CTR1, and ETR1 (a jasmonate receptor, ethylene receptor, and ethylene response mediator, respectively).

Overrepresented subnetworks downregulated in both teosinte lines limited to only 9 (S3 File). The common subnetworks were all related to light, photosynthesis, or carbon cycles. Several more downregulated subnetworks unique to Ames 21789 were related to photosynthesis, such as regulation of photosynthetic acclimation and the binding partners of the photosystem II reaction center. Varying subnetworks regulating plant growth were uniquely downregulated in Ames 21812, such as regulation of greening, growth rate, hypocotyl growth, and ripening, among others.

A small group of significant overrepresented subnetworks consisted of 4 which were common to both lines involved up-regulation of plant defense and the defense response, lignification (Fig 3), and systemic acquired resistance (Table 7). All common and unique overrepresented subnetworks were upregulated in weed-stressed plants, except for regulation of shoot growth, which was uniquely down regulated in Ames 21812 and interestingly maintained biomass/branches, unlike Ames 21789.

## Differences in teosinte transcriptomic response visualized by Mapman

Differentially expressed gene pathways were visualized by Mapman, a simple program utilized to visualize differences in various gene ontologies. The Metabolism Overview mapping scheme identifies differences and similarities between the two teosinte lines using DEGs (Figs 4 and 5). In the interactive software program, the small red or green squares indicate individual DEGs which are either down-regulated (red), or up-regulated (green). These squares may be selected to reveal specific information regarding each DEG, such as log fold change, full gene name, function, etc. Several differences (i.e. fermentation, TCA), as well as similarities (i.e. tetrapyrrole) between varieties response to weeds were indicated.

## Differences and similarities between maize and teosinte response to weeds

Over-represented gene ontologies observed among both teosinte varieties associated with weed interference were compared between a previously published list from the response of maize to weeds [28]. Among the 190 over-represented gene and sub-network ontologies observed among up-regulated genes in both teosinte varieties, 24 were also observed among

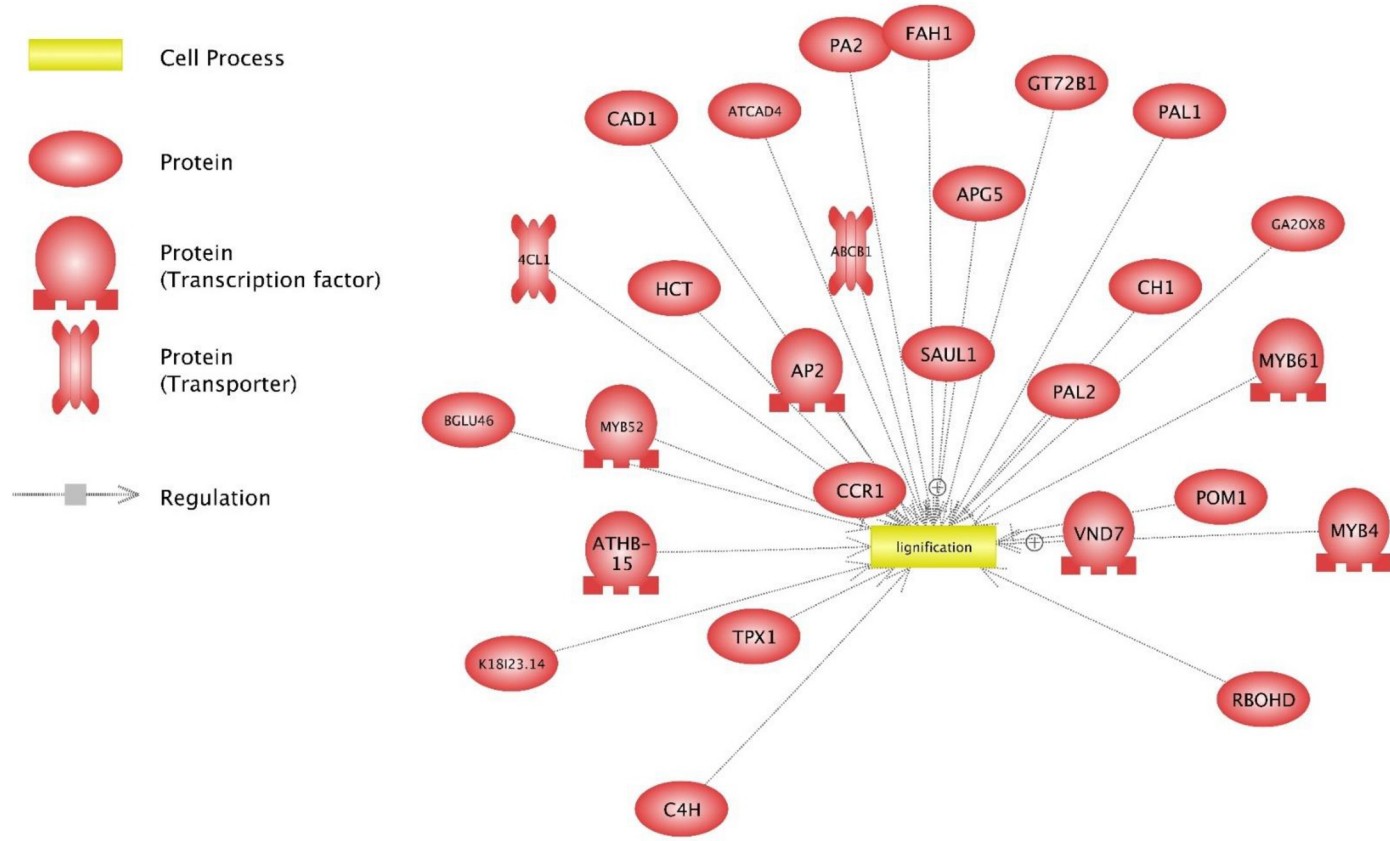

**Fig 3. Proteins/chemicals regulating cellular processes of lignification.** Illustration of up-regulated significant genes in Ames 21789 and their interaction with the lignification network. Visualized using Pathway Studio.

the 42 over-represented ontologies in maize (Fig 6). Likewise, of the 83 over-represented ontologies observed in genes that were repressed by weed interference in teosinte, 19 of the 47 were also observed in maize in both years of the study. Commonly over-represented pathways among the down-regulated genes, as well as a subset of the over-represented among genes up-regulated by weed interference, are shown in Table 8. Notable similarities include the reduction in photosynthetic processes, and up-regulation of salicylic acid, jasmonic acid and defense responses in both maize and teosinte. However, there were also a large number of over-represented ontologies that were unique to either species response to weed interference (Table 9: full list in S3 File). These identify processes that may have been selected in maize during domestication–potentially as a primarily intercropped species commonly grown alongside beans and squash [32]. Notable among these appears to be greater secondary metabolic processes including flavonoid/phenylpropanoid metabolism processes observed in the teosinte response to weeds but absent from maize. Additionally, maize seems to have gained more intense phytochrome and hormone responses that could affect growth and physiology in the presence of weeds. Indeed, commonly observed light signaling processes associated with plant-plant interactions appears to be repressed in teosinte.

## Discussion

Crop wild species have served as genetic resources for successful crop improvement in a number of instances [33–36]. Multiple research paths are underway to improve maize using

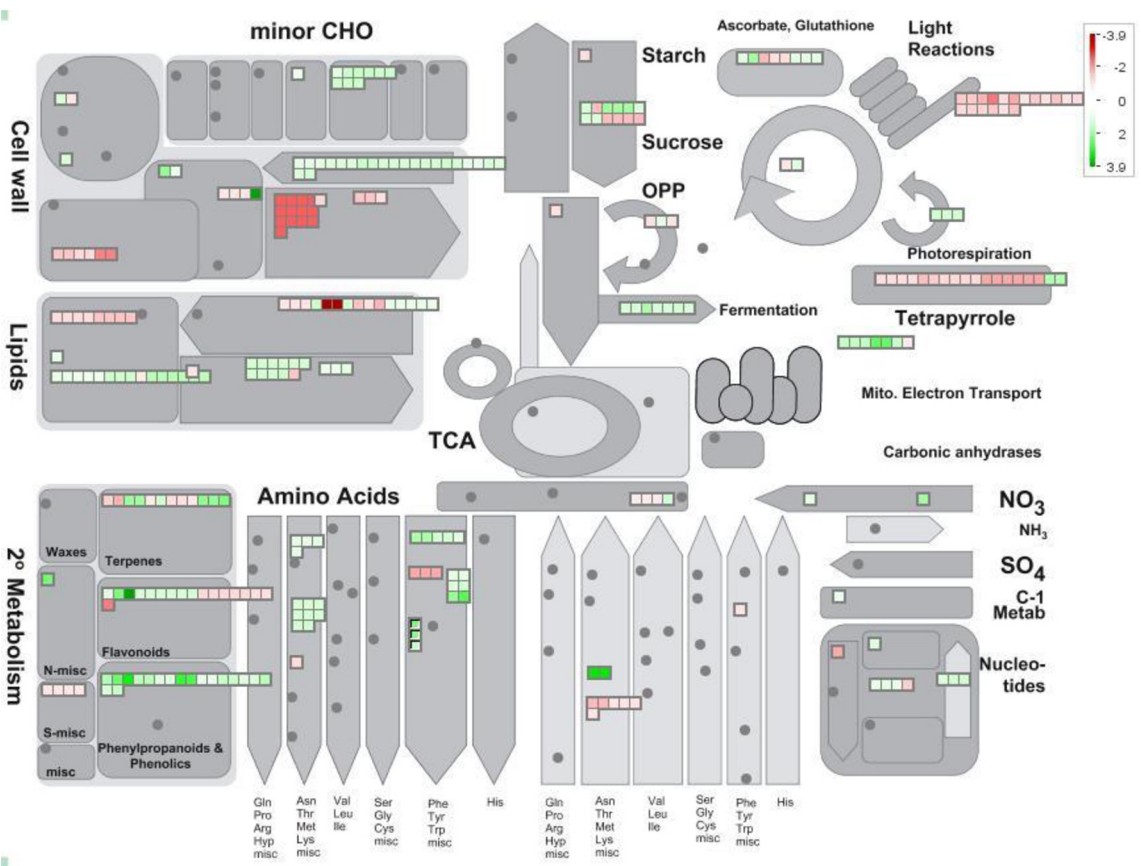

**Fig 4. Metabolism overview in teosinte line Ames 21789 visualized by Mapman.** Squares in red (down-regulated) or green (up-regulated) indicate individual DEGs categorized as shown.

genetics or other beneficial above and belowground attributes of teosinte [22, 23, 37, 38]. For example, Burton et al. [37], investigated root architecture in teosinte for phenotypic diversity in hopes of improving stress tolerance in maize. In addition, differences in pest resistance between maize and teosinte was evaluated by De Lange et al. [23]. As more information about the role in plant health for species-associated below ground biota, researchers are investigating the differences in biota attributes among differing climates where teosinte is grown [22, 38]. Both of these examples provide information that could be useful for improving maize.

Atkinson and Urwin [39] stated plants respond differently to multiple stresses than they do to individual stresses, and that response to a particular stress may depend totally upon the specific environmental condition the plant finds itself in when stressed. Signaling pathways and molecular mechanisms involved in multi stress responses may compound or reduce various pathways and effects. Indeed, the different weed populations and densities and different growing conditions between years likely contributed significantly to variation in both teosinte growth and transcriptome responses. Thus, greenhouse and controlled studies may give a direction or general idea, but the complexities generated by different environments and varietal responses is a confounding effect in the quest for determining an efficient means to mediate yield loss due to weed presence. Information abounds regarding individual gene response to heat, salt, water, weed stress and other components in highly controlled situations. Investigations of the interactive effects genes impart upon one another in a plant system in its natural habitat while under weed stress is an under-represented area of research.

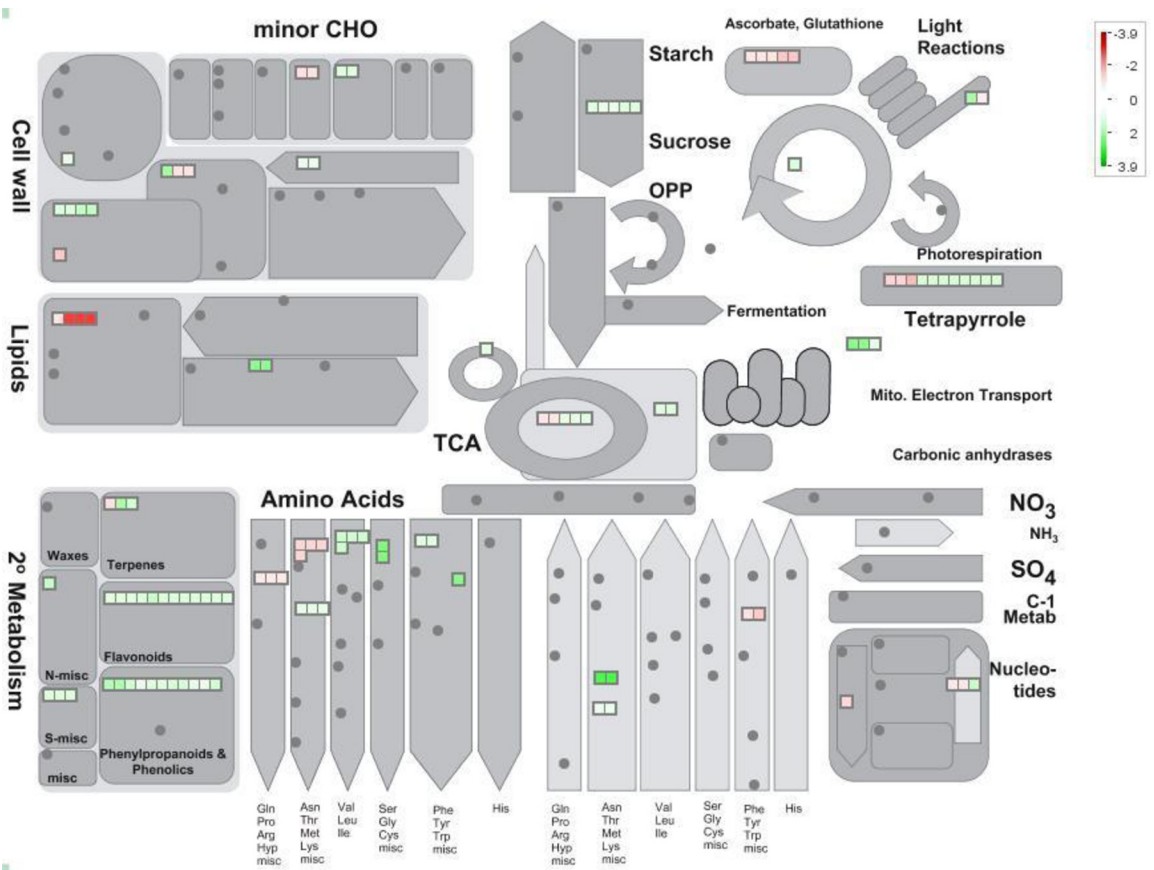

**Fig 5. Metabolism overview in teosinte line Ames 21812 visualized by Mapman.** Squares in red (down-regulated) or green (up-regulated) indicate individual DEGs categorized as shown.

Numerous studies utilizing quantitative trait loci (QTL), single nucleotide polymorphisms (SNPs), simple sequence repeats (SSRs), and mapping methods to investigate domestication events and the genes involved in the domestication of maize have garnered a large amount of data. However, few, if any, studies utilizing RNA sequencing to evaluate gene expression differences between stressed and unstressed teosinte in field settings have taken place. This study is novel in the use of RNA sequencing in a field environment evaluating weed effects in a naturally fluctuating environment on gene expression in teosinte. Swanson-Wagner et al. [40] suggested gene content variation found in large gene families allows for a core genome shared by all members of a species, and a non-core genome, which would fluctuate and create phenotypic diversity. This would result in "overall" responses to stresses, but potentially no one particular gene or small sets of genes controlling the response.

There were marked difference in both phenology and transcriptome response to weed stress between teosinte lines, as demonstrated by the differences in plant height, branch number, biomass per plant, and the number of unique transcriptomic response categories. Teosinte down-regulated several gene ontologies related to photosystems and chloroplasts in response to weed pressure, and this was also observed in maize [28]. Likewise, both species showed general up-regulation of biotic stress responses–specifically the induction of SA signaling responses. Oxidative stress responses, in which flavonoid responses likely play a protective role, have been implicated as playing a role in early response of maize to weeds [41].

# Pathways and processes up with weeds

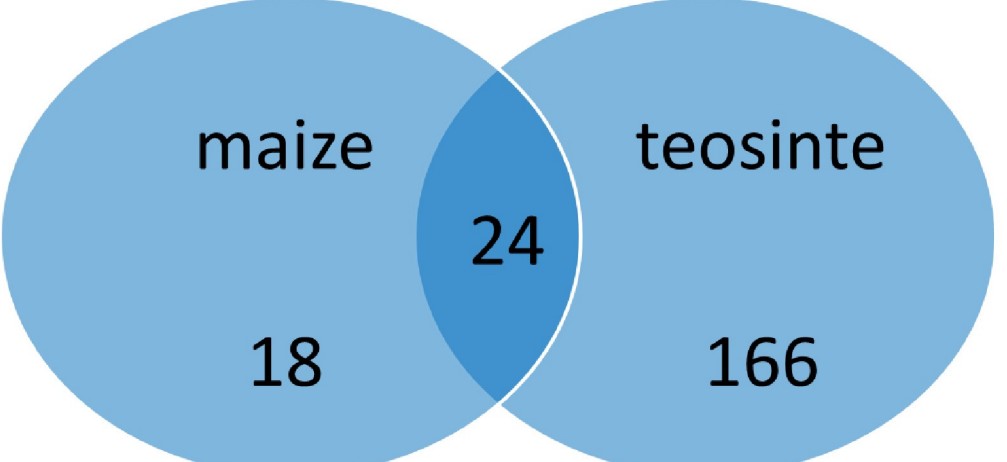

# Pathways and processes down with weeds

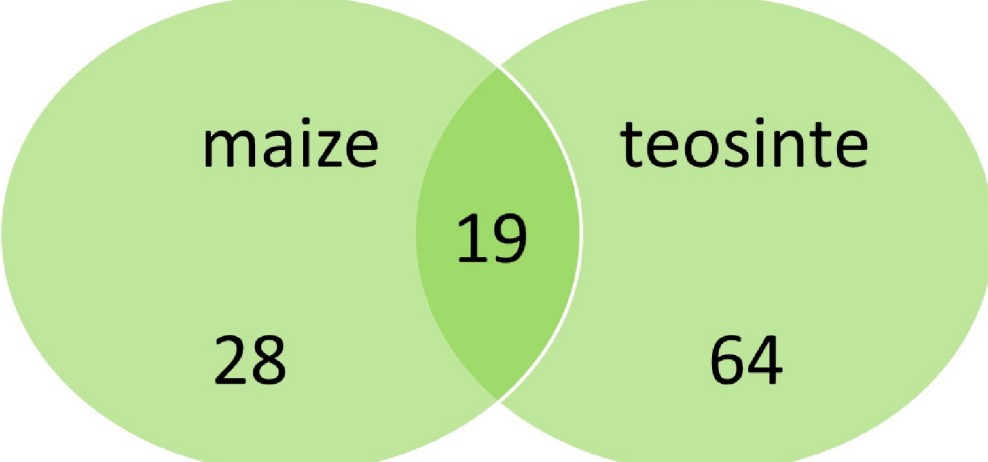

**Fig 6. Venn diagram indicating similarities and overlaps among gene ontologies.** "Up with weeds" and "down with weeds" refer ontologies over-represented among genes that were up-regulated by weed interference or down-regulated by weed interference respectively in maize and/or teosinte as indicated.

Further investigation is needed over growing seasons and multiple teosinte lines to evaluate the function of UDP glucosyl- and glycosyltransferases, quercetin glucosyltransferase, and jasmonic acid signaling in weed response in teosinte and maize. Likewise, only the nodulin *MtN21* gene was up-regulated in both lines under weed stress. Thus, it might serve as a tool for identifying signaling networks involved in interference-sensing/response of teosinte, as it likely contains species-specific regulatory elements required for up-regulation in response to weeds. Other coordinately regulated gene sets could also serve to identify such elements but may be cultivar specific. Finally, the variation in response between these two lines, and mapping of the genes controlling these differences, may help shed some light on the interference-regulated growth and developmental processes of maize, and provide insights into the selection pressures that have impacted maize plant architecture under inter- and intra-species interference.

**Table 8. Over-represented gene ontologies common between maize and teosinte.**

| Up during weed interference in both maize and teosinte | Down during weed interference in both maize and teosinte |
|---|---|
| calmodulin binding | Apoplast |
| cellular response to phosphate starvation | aromatic amino acid family biosynthetic process |
| defense response | Chloroplast |
| defense response to bacterium | chloroplast envelope |
| defense response to fungus | chloroplast organization |
| detection of biotic stimulus | chloroplast relocation |
| hyperosmotic salinity response | chloroplast stroma |
| iron ion binding | iron-sulfur cluster assembly |
| jasmonic acid mediated signaling pathway | ncRNA metabolic process |
| Jasmonic Acid Signaling | Nucleoid |
| plant-type hypersensitive response | ovule development |
| protein targeting to membrane | plastid chromosome |
| regulation of hydrogen peroxide metabolic process | protein targeting to chloroplast |
| regulation of plant-type hypersensitive response | Proteins/Chemicals Regulating Cell Processes of chloroplast organization and biogenesis |
| response to bacterium | rRNA binding |
| response to chitin | rRNA processing |
| response to hypoxia | Thylakoid |
| response to karrikin | thylakoid membrane organization |
| response to water deprivation | transcription from plastid promoter |
| salicylic acid biosynthetic process | |
| salicylic acid mediated signaling pathway | |
| sequence-specific DNA binding | |
| signal transduction | |
| systemic acquired resistance, SA mediated signaling pathway | |

**Table 9. Subset of over-represented gene ontologies unique to either maize or teosinte.** In these subsets, groups of ontologies that were unique by name, but related in function to ontologies present in the alternate species were avoided.

| Up during weed interference unique to maize | Down during weed interference unique to maize |
|---|---|
| phytochrome Signaling | Binding Partners of ribosome |
| abscisic acid mediated signaling pathway | heme binding |
| response to ethylene stimulus | peroxidase activity |
| response to gibberellin stimulus | PCRCP of phototropism |
| response to osmotic stress | regulation of meristem growth |
| **Up during weed interference unique to teosinte** | **Down during weed interference to teosinte** |
| amino acid transmembrane transport | response to blue light |
| lignin biosynthetic process | response to far red light |
| quercetin 3-O-glucosyltransferase activity | unsaturated fatty acid biosynthetic process |
| flavonoid biosynthetic process | chloroplast thylakoid |
| phenylpropanoid metabolic process | Binding Partners of light-harvesting complex |

## Supporting information

**S1 Table. Table of the number of reads and mapping success of the various libraries.**
(DOCX)

**S1 File. Assembled fasta file from the guided assembly.**
(FASTA)

**S2 File. Annotation and expression data (FPKM) for all genes in all samples.** Additionally,
False Discovery Rate (FDR) values are provided for weedy verses control in both Ames 21789
and Ames 21812.
(XLSX)

**S3 File. Gene set enrichment analysis from each individual comparison between weedy
verses non-weedy teosinte.** Additionally, overlap between the over-represented ontologies for
both cultivars as well as ontologies that are unique to each cultivar is provided. Likewise, the
common over-represented gene ontologies from maize growing with verses without weeds as
extracted from [28] are provided as well as the overlap observed among over-represented
ontologies from both maize and teosinte.
(XLSX)

## Author Contributions

**Conceptualization:** S. A. Bruggeman, D. P. Horvath, S. A. Clay.

**Data curation:** S. A. Bruggeman, D. P. Horvath.

**Formal analysis:** S. A. Bruggeman, D. P. Horvath, S. A. Clay.

**Funding acquisition:** S. A. Bruggeman, D. P. Horvath, S. A. Clay.

**Investigation:** S. A. Bruggeman, D. P. Horvath, S. A. Clay.

**Methodology:** S. A. Bruggeman, D. P. Horvath, S. A. Clay.

**Project administration:** S. A. Bruggeman, D. P. Horvath, S. A. Clay.

**Resources:** S. A. Bruggeman, D. P. Horvath, S. A. Clay.

**Software:** S. A. Bruggeman, D. P. Horvath.

**Supervision:** S. A. Bruggeman, D. P. Horvath, A. Y. Fennell, J. L. Gonzalez-Hernandez, S. A. Clay.

**Validation:** S. A. Bruggeman, D. P. Horvath, S. A. Clay.

**Visualization:** S. A. Bruggeman, D. P. Horvath.

**Writing – original draft:** S. A. Bruggeman, D. P. Horvath.

**Writing – review & editing:** S. A. Bruggeman, D. P. Horvath, A. Y. Fennell, J. L. Gonzalez-Hernandez, S. A. Clay.

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
