## [Decision Letter · Decision Letter 0]

21 Apr 2020

PONE-D-20-04184

Teosinte (Zea mays ssp parviglumis) growth and transcriptomic response to weed stress identifies similarities and differences between varieties and with modern maize varieties.

PLOS ONE

Dear Dr. Horvath,

Thank you for submitting your manuscript to PLOS ONE. After careful consideration, we feel that it has merit but does not fully meet PLOS ONE’s publication criteria as it currently stands. Therefore, we invite you to submit a revised version of the manuscript that addresses the points raised during the review process.

We would appreciate receiving your revised manuscript by Jun 05 2020 11:59PM. To enhance the reproducibility of your results, we recommend that if applicable you deposit your laboratory protocols in protocols.io, where a protocol can be assigned its own identifier (DOI) such that it can be cited independently in the future. For instructions see: http://journals.plos.org/plosone/s/submission-guidelines#loc-laboratory-protocols

We look forward to receiving your revised manuscript.

Kind regards,

Anil Kumar Singh, Ph.D.

Academic Editor

PLOS ONE

Reviewers' comments:

Reviewer's Responses to Questions

**Comments to the Author**

1. Is the manuscript technically sound, and do the data support the conclusions?

Reviewer #1: Yes

Reviewer #2: Yes

2. Has the statistical analysis been performed appropriately and rigorously? 

Reviewer #1: Yes

Reviewer #2: Yes

3. Have the authors made all data underlying the findings in their manuscript fully available?

Reviewer #1: Yes

Reviewer #2: Yes

4. Is the manuscript presented in an intelligible fashion and written in standard English?

Reviewer #1: No

Reviewer #2: Yes

5. Review Comments to the Author

Reviewer #1: General comment:

The manuscript entitled "Teosinte (Zea mays ssp parviglumis) growth and transcriptomic response to weed stress identifies similarities and differences between varieties and with modern maize varieties.." has been submitted by Authors with the aim to evaluate the transcriptomic responses of teosinte (Zea mays ssp parviglumis), that is an ancestor of domesticated maize, to weed presence over two growing seasons. During this study morphological and transcriptomic responses of two teosinte lines i.e. Ames 21812 and Ames 21789, grown with and without weed presence for 6 weeks in Aurora, South Dakota, USA, were compared. During the treatments both teosinte lines Ames 21812 and Ames 21789 shows contrasting characteristics with and without weed presence making a good candidate for determining and characterizing the genes underlying these responses in modern maize varieties. Biomass accumulation decreased in both lines under weed presence. Downregulation in the networks related to light, photosynthesis, and carbon cycles was observed. Many unique response networks (like aging, response to chitin) and gene sets were noticed in each line. Transcriptome based analysis in teosinte lines indicated the upregulation of three gene ontologies in weed presence were jasmonic acid response/signaling, UDP-glucosyl and glucuronyltransferases, and quercetin glucosyltransferase. Based on above findings authors suggest that there might be varietal responses to weed stress that may be used to manipulate the modern maize varieties. Authors have selected very much important question and performed experiment and analyzed data using recent tools/approaches. However, the following points may be clarified before its acceptance.

Abstract: It needs to be advanced.

Like//- These observations suggest significant differences in response to interference exist in the progenitors of maize, and determination of the genes underlying these responses should be characterized in modern maize varieties.

Query :- Authors may elaborate the type of interference is being described.

Introduction: This section is well written having information collected from literature. However, some points need to be addressed.

//43 the domestication of crops [1-11]. Research has determined modern day maize diverged

44 from its wild progenitor Zea mays ssp. parviglumis (teosinte) approximately 9000 years

45 ago [12-14].

Query?? :- Authors may mention either Zea mays ssp. parviglumis (teosinte) or Teosinte (Zea mays ssp. parviglumis) in whole MS

Materials and methods: This section is also well written having sequential steps followed for data collection and analysis. However, some points need to be advanced.

//250 branch number. September branch number in teosinte was negatively influenced in weed

251 stressed plants both years.

//255 Teosinte harvest biomass followed decreasing trends of September branch

256 number and plant height.

Query??:- Authors may advance the sentence “September branch number in teosinte---"more appropriately.

//155 Table 1. Seed quality values for teosinte lines evaluated in 2014 at the South Dakota

156 Research Farm, Aurora, South Dakota. Seed data based on Flint-Garcia et al., 2009.

Line Prot Fat Fiber Ash Carb

Query??:- Authors need to mentioned full form of abbreviations used in Table-1 like: Prot and Carb

//205 following a Fisher’s Exact test for over-representation. MAPMAN was also used in a

206 similar manor but was limited to analysis of only DEGs to visualize gene expression

Query??:- Authors may look this sentence and may write appropriately about “similar manor”

Results: Obtained results have been elaborated well. However, following points need to be addressed.

//225 densities averaged from 85 (+2) plants/meter2 in Ames 21789 to 285 (+16) in the Ames

226 21812 weedy plots, and weed biomass was similar in weedy plots between teosinte lines

227 (averaged from 820 (+385) to 900 (+426) g/m2)) (Table 5) .

Query??:- Authors need to check the units and make it uniform in whole MS

//Table 3. Monthly temperature and precipitation averages with 30-year normal over 2013

260 and 2014 growing seasons.

Query??:- Authors need to mentioned full form of abbreviations used in Table-3 & 4 like: Precip

Query??:- Table 5: Authors need to check the unit of weed density: it may be plantsm2 or plants/ m2

// Page 22. Only one DEG was shared between the two lines, GRMZM2G114751, a nodulin MtN21 family protein, involved in transmembrane transport, which was up-regulated in both Ames 21789 and Ames 21812 weed-stressed plants in comparison to their weed-free counterparts (2.4 and 4.6-fold change, respectively)

Query??:- Authors may elaborate more about: “Only one DEG was shared between the two lines, GRMZM2G114751, a nodulin MtN21 family protein” indicating how this study is going to enrich the existing information?

//Table 7. Subnetworks overexpressed in significant genes in weed-stressed teosinte plants compared with weed-free plants.

Query??:- Authors may refine the legend of Table 7.

// Page 32. Several differences (i.e. fermentation, TCA), as well as similarities (i.e. tetrapyrrole)

Query??:- Incomplete sentence

// Page 34. These identify processes that may have been selected in maize during domestication –

potentially as a primarily intercropped species commonly grown alongside beans and squash.

Query??:- Authors may refine the sentence.

Discussion: This section indicates the discussion related to silent findings. However, following points need to be considered.

//Page 38: For example, [37], investigated root architecture in-----

//Page 38: In addition, differences in pest resistance between maize and teosinte was evaluated by [23].

Query??:- In my opinion citation in above sentences may be followed as authors have cited [at Page 33: “Atkinson and Urwin [39] stated plants respond differently to multiple stresses” and “Swanson-Wagner et al. [40] suggested gene content variation”.

//Page 39: Information abounds regarding individual gene response to heat, salt, water, and other components in highly controlled situations. The interactive effects genes impart upon one another in a plant system in its natural habitat while under weed stress is a completely different area of research.

Query??:- Authors may add the probable reasons for stating weed stress as a completely different area of research

//Page 40: Likewise, the up-regulated nodulin MtN21 gene may serve as a tool for identifying signaling networks involved in interference-sensing/response of teosinte.

Query??:- As per S2 File many there are many DEGs. Why authors emphasizing only MtN21 gene for future study. Also, MtN21 gene need to be in italics.

References

//Reference 10. Hufford MB, Lubinksy P, Pyajarvi T, Devengenzo MT, Ellestrand NC, Ross-

Ibarra J. The genomic signature of crop-wild introgression in maize. PLoS Genet.

2013;9: 10.1371/annotation/2eef7b5b-29b2-412f-8472-8fd7f9bd65ab

doi:10.1371/annotation/2eef7b5b-29b2-412f-8472-8fd7f9bd65ab

Query??:- Authors need to check and correct the Reference 10

Overall, authors may add biological significance of data, elaborate the content and queries to make the manuscript conclusive as well as appealing for more readership.

Reviewer #2: Line 88. field conditions and to identify similarities and differences to (in) the transcriptomic response.

Query 1. How many times weeding was done to maintain the field weed-free, as weeds may come out within one week.

Query 2. As the weed species density varies in two years and biomass was similar, that hints that the weed species were different in two years, it would be better if authors can specify the weed species in each year, its number not the overall weed species, its density, and biomass, As each weed response can be different.

Query 3: Can Author justify a decrease in branch number in Ames 21812 in response to weed stress, from July (2.2) to 1.4 ( September).

Query 4. The authors did not mention the exact number of plants used for statistical analysis and phenotype parameters. If just 6 or 4 plants in 2013 and 2014 grown in a 3mX3m plot, it would be challenging to conclude the parameters.

Query 5. Have authors seen any difference in root length or diameter in response to weed stress, as root would be the most prone part to compete with weeds for nutrients?

6. PLOS authors have the option to publish the peer review history of their article (what does this mean?). If published, this will include your full peer review and any attached files.

Reviewer #1: Yes: Dev Mani Pandey

Reviewer #2: Yes: Ritesh Kumar

---

## [Author Response · Author response to Decision Letter 0]

22 May 2020

Review Comments to the Author

Reviewer #1: General comment:

The manuscript entitled "Teosinte (Zea mays ssp parviglumis) growth and transcriptomic response to weed stress identifies similarities and differences between varieties and with modern maize varieties.." has been submitted by Authors with the aim to evaluate the transcriptomic responses of teosinte (Zea mays ssp parviglumis), that is an ancestor of domesticated maize, to weed presence over two growing seasons. During this study morphological and transcriptomic responses of two teosinte lines i.e. Ames 21812 and Ames 21789, grown with and without weed presence for 6 weeks in Aurora, South Dakota, USA, were compared. During the treatments both teosinte lines Ames 21812 and Ames 21789 shows contrasting characteristics with and without weed presence making a good candidate for determining and characterizing the genes underlying these responses in modern maize varieties. Biomass accumulation decreased in both lines under weed presence. Downregulation in the networks related to light, photosynthesis, and carbon cycles was observed. Many unique response networks (like aging, response to chitin) and gene sets were noticed in each line. Transcriptome based analysis in teosinte lines indicated the upregulation of three gene ontologies in weed presence were jasmonic acid response/signaling, UDP-glucosyl and glucuronyltransferases, and quercetin glucosyltransferase. Based on above findings authors suggest that there might be varietal responses to weed stress that may be used to manipulate the modern maize varieties. Authors have selected very much important question and performed experiment and analyzed data using recent tools/approaches. However, the following points may be clarified before its acceptance.

Abstract: It needs to be advanced.

Like//- These observations suggest significant differences in response to interference exist in the progenitors of maize, and determination of the genes underlying these responses should be characterized in modern maize varieties.

Query :- Authors may elaborate the type of interference is being described.

Response: Line 25: “interference” has been changed to “weed stress” 

Introduction: This section is well written having information collected from literature. However, some points need to be addressed.

//43 the domestication of crops [1-11]. Research has determined modern day maize diverged

44 from its wild progenitor Zea mays ssp. parviglumis (teosinte) approximately 9000 years

45 ago [12-14].

Query?? :- Authors may mention either Zea mays ssp. parviglumis (teosinte) or Teosinte (Zea mays ssp. parviglumis) in whole MS

Response: Authors have corrected the manuscript so that “teosinte” is used throughout the manuscript after the first mention, where “teosinte (Zea mays ssp. parviglumis)” is utilized, per instructions to authors.

Materials and methods: This section is also well written having sequential steps followed for data collection and analysis. However, some points need to be advanced.

//250 branch number. September branch number in teosinte was negatively influenced in weed

251 stressed plants both years.

//255 Teosinte harvest biomass followed decreasing trends of September branch

256 number and plant height.

Query??:- Authors may advance the sentence “September branch number in teosinte---"more appropriately.

Response: Authors have changed “was negatively influenced” to “decreased in weed stressed plants in comparison to weed-free plants.”

//155 Table 1. Seed quality values for teosinte lines evaluated in 2014 at the South Dakota

156 Research Farm, Aurora, South Dakota. Seed data based on Flint-Garcia et al., 2009.

Line Prot Fat Fiber Ash Carb

Query??:- Authors need to mentioned full form of abbreviations used in Table-1 like: Prot and Carb

Response: Authors have included abbreviations in the table heading for Table 1.

//205 following a Fisher’s Exact test for over-representation. MAPMAN was also used in a

206 similar manor but was limited to analysis of only DEGs to visualize gene expression

Query??:- Authors may look this sentence and may write appropriately about “similar manor”

Response: The Mapman method has been clarified by rewriting to include “create hierarchical and non-redundant gene ontologies through analysis of DEGs.”

Results: Obtained results have been elaborated well. However, following points need to be addressed.

//225 densities averaged from 85 (+2) plants/meter2 in Ames 21789 to 285 (+16) in the Ames

226 21812 weedy plots, and weed biomass was similar in weedy plots between teosinte lines

227 (averaged from 820 (+385) to 900 (+426) g/m2)) (Table 5) .

Query??:- Authors need to check the units and make it uniform in whole MS

Response: Authors corrected all g/m or g/plant to read gm/m and gm/plant, etc.

//Table 3. Monthly temperature and precipitation averages with 30-year normal over 2013

260 and 2014 growing seasons.

Query??:- Authors need to mentioned full form of abbreviations used in Table-3 & 4 like: Precip

Query??:- Table 5: Authors need to check the unit of weed density: it may be plantsm2 or plants/ m2

Response: Abbreviations have been added to table headings, and plantsm2 has been corrected to plants/m2.

// Page 22. Only one DEG was shared between the two lines, GRMZM2G114751, a nodulin MtN21 family protein, involved in transmembrane transport, which was up-regulated in both Ames 21789 and Ames 21812 weed-stressed plants in comparison to their weed-free counterparts (2.4 and 4.6-fold change, respectively)

Query??:- Authors may elaborate more about: “Only one DEG was shared between the two lines, GRMZM2G114751, a nodulin MtN21 family protein” indicating how this study is going to enrich the existing information?

The implications for this particular gene to enrich existing information is unclear. We simply are reporting the results. However, the consistent differential expression does indicate that this gene might serve as a source of weed-responsive regulatory elements as was noted in the discussion.

//Table 7. Subnetworks overexpressed in significant genes in weed-stressed teosinte plants compared with weed-free plants.

Query??:- Authors may refine the legend of Table 7.

Response: To clarify, “overexpressed in significant genes” has been changed to “overexpressed through significant genes.”

// Page 32. Several differences (i.e. fermentation, TCA), as well as similarities (i.e. tetrapyrrole)

Query??:- Incomplete sentence

Sentence now reads “Several differences (i.e. fermentation, TCA), as well as similarities (i.e. tetrapyrrole) between varieties response to weeds were indicated.”

// Page 34. These identify processes that may have been selected in maize during domestication –

potentially as a primarily intercropped species commonly grown alongside beans and squash.

Query??:- Authors may refine the sentence.

Response: Authors have rephrased the sentence to read “provide insights into the selection pressures that have no doubt impacted maize plant architecture under inter- and intra-species interference.”

Discussion: This section indicates the discussion related to silent findings. However, following points need to be considered.

//Page 38: For example, [37], investigated root architecture in-----

//Page 38: In addition, differences in pest resistance between maize and teosinte was evaluated by [23].

Query??:- In my opinion citation in above sentences may be followed as authors have cited [at Page 33: “Atkinson and Urwin [39] stated plants respond differently to multiple stresses” and “Swanson-Wagner et al. [40] suggested gene content variation”.

Response: The authors agree with the suggestion, and have corrected as such.

//Page 39: Information abounds regarding individual gene response to heat, salt, water, and other components in highly controlled situations. The interactive effects genes impart upon one another in a plant system in its natural habitat while under weed stress is a completely different area of research.

Query??:- Authors may add the probable reasons for stating weed stress as a completely different area of research

We agree and the sentences now read: “Information abounds regarding individual gene response to heat, salt, water, weed stress and other components in highly controlled situations. Investigations of the interactive effects genes impart upon one another in a plant system in its natural habitat while under weed stress is an under-represented area of research.”

//Page 40: Likewise, the up-regulated nodulin MtN21 gene may serve as a tool for identifying signaling networks involved in interference-sensing/response of teosinte.

Query??:- As per S2 File many there are many DEGs. Why authors emphasizing only MtN21 gene for future study. Also, MtN21 gene need to be in italics.

Response: As previously mentioned in the Results section, MtN21 was the only _shared_ DEG between the two lines when placed under weed stress in field conditions. Authors have added that justification to indicate as such. Other genes that were not differential in both lines were included in the gene set enrichment analysis which are better at identifying common responses to weed stress. 

References

//Reference 10. Hufford MB, Lubinksy P, Pyajarvi T, Devengenzo MT, Ellestrand NC, Ross-

Ibarra J. The genomic signature of crop-wild introgression in maize. PLoS Genet.

2013;9: 10.1371/annotation/2eef7b5b-29b2-412f-8472-8fd7f9bd65ab

doi:10.1371/annotation/2eef7b5b-29b2-412f-8472-8fd7f9bd65ab

Query??:- Authors need to check and correct the Reference 10

Response: Authors thank the reviewer for finding our error. The correction to the name Lubinsky has been made.

Overall, authors may add biological significance of data, elaborate the content and queries to make the manuscript conclusive as well as appealing for more readership.

Reviewer #2: Line 88. field conditions and to identify similarities and differences to (in) the transcriptomic response.

Query 1. How many times weeding was done to maintain the field weed-free, as weeds may come out within one week.

Response: M&M have been amended to include: “…were maintained weed free by hand hoeing and weeding approximately once every 7-10 days during the growing season…”

Query 2. As the weed species density varies in two years and biomass was similar, that hints that the weed species were different in two years, it would be better if authors can specify the weed species in each year, its number not the overall weed species, its density, and biomass, As each weed response can be different.

Response: Unfortunately, no species specific data was taken at the time. Visual assessment of plot photographs indicate the majority of biomass appears to be from broadleaf species such as kochia and pigweed in 2013 and velvetleaf and grasses predominated in 2014 as indicated in the results section. We now note in the discussion that these differences as well as climactic differences between the years could explain the year to year differences we observed. However, it should be noted that despite these year to year differences, the weeds still clearly had significant impacts of the phenology and transcriptome responses of teosinte in both years. 

Query 3: Can Author justify a decrease in branch number in Ames 21812 in response to weed stress, from July (2.2) to 1.4 (September).

Response: As in maize, lower leaves may desiccate and fall off as the season progresses. While the weed-stressed plants may have been able to support more branches in the early season due to more resources being available, competition for more resources over the length of the season may have led to the teosinte no longer being able to maintain the previous number of branches, leading to allocation of resources shifting to the bigger, sturdier main leaves and subsequent loss of the weaker branches later in the season. 

Query 4. The authors did not mention the exact number of plants used for statistical analysis and phenotype parameters. If just 6 or 4 plants in 2013 and 2014 grown in a 3mX3m plot, it would be challenging to conclude the parameters.

Response: At the time, researchers were only allowed 100-350 seeds from each teosinte line from GRIN (US National Plant Germplasm storage). In order to perform 2 years of research with two treatments and multiple plots, the total number of seeds planted per year were minimal. However, we collected data from all plants in each plot (resulting in data from 16 to 24 plants per treatment depending on the year. 

Query 5. Have authors seen any difference in root length or diameter in response to weed stress, as root would be the most prone part to compete with weeds for nutrients?

Response: Root lengths were not evaluated in this study. However, we have initiated investigations into the transcriptome response of corn roots to weeds under controlled conditions- look for data from these studies at the ASPB meeting in July.

---

## [Decision Letter · Decision Letter 1]

18 Jun 2020

PONE-D-20-04184R1

Teosinte (Zea mays ssp parviglumis) growth and transcriptomic response to weed stress identifies similarities and differences between varieties and with modern maize varieties.

PLOS ONE

Dear Dr. Horvath,

Thank you for submitting your manuscript to PLOS ONE. After careful consideration, we feel that it has merit but does not fully meet PLOS ONE’s publication criteria as it currently stands. Therefore, we invite you to submit a revised version of the manuscript that addresses the points raised during the review process.

Authors are advised to make the minor corrections/changes as suggested by both the reviewers.

We look forward to receiving your revised manuscript.

Kind regards,

Anil Kumar Singh, Ph.D.

Academic Editor

PLOS ONE

Reviewers' comments:

Reviewer's Responses to Questions

**Comments to the Author**

1. If the authors have adequately addressed your comments raised in a previous round of review and you feel that this manuscript is now acceptable for publication, you may indicate that here to bypass the “Comments to the Author” section, enter your conflict of interest statement in the “Confidential to Editor” section, and submit your "Accept" recommendation.

Reviewer #1: All comments have been addressed

Reviewer #2: All comments have been addressed

2. Is the manuscript technically sound, and do the data support the conclusions?

Reviewer #1: Yes

Reviewer #2: Yes

3. Has the statistical analysis been performed appropriately and rigorously? 

Reviewer #1: Yes

Reviewer #2: Yes

4. Have the authors made all data underlying the findings in their manuscript fully available?

Reviewer #1: Yes

Reviewer #2: Yes

5. Is the manuscript presented in an intelligible fashion and written in standard English?

Reviewer #1: Yes

Reviewer #2: Yes

6. Review Comments to the Author

Reviewer #1: General comment:

The manuscript entitled "Teosinte (Zea mays ssp parviglumis) growth and transcriptomic response to weed stress identifies similarities and differences between varieties and with modern maize varieties.." has been revised and re-submitted by Authors with the aim to evaluate the transcriptomic responses of teosinte (Zea mays ssp parviglumis), that is an ancestor of domesticated maize, to weed presence over two growing seasons.

Substantial modifications have been done by Authors to make the manuscript more conclusive and appealing as well as upto acceptable form.

Further, refining of the last sentence of Introduction as well as correction in Reference 10 may be done in galley proof.

Reviewer #2: The authors have addressed all the previous comments and integrated a few remarks in the revised version of the manuscript. But answers to some comments are just not queries, and those can be included in the discussion, which can address these arisen questions.

7. PLOS authors have the option to publish the peer review history of their article (what does this mean?). If published, this will include your full peer review and any attached files.

Reviewer #1: Yes: Dev Mani Pandey

Reviewer #2: Yes: Ritesh Kumar

---

## [Author Response · Author response to Decision Letter 1]

23 Jul 2020

Response to reviews II

Reviewer #1: General comment:

The manuscript entitled "Teosinte (Zea mays ssp parviglumis) growth and transcriptomic response to weed stress identifies similarities and differences between varieties and with modern maize varieties.." has been revised and re-submitted by Authors with the aim to evaluate the transcriptomic responses of teosinte (Zea mays ssp parviglumis), that is an ancestor of domesticated maize, to weed presence over two growing seasons.

Substantial modifications have been done by Authors to make the manuscript more conclusive and appealing as well as up to acceptable form.

Further, refining of the last sentence of Introduction as well as correction in Reference 10 may be done in galley proof.

Fixed the typos in both sentence and reference #10

Reviewer #2: The authors have addressed all the previous comments and integrated a few remarks in the revised version of the manuscript. But answers to some comments are just not queries, and those can be included in the discussion, which can address these arisen questions.

I am not sure what queries were not addressed. We did further refine our discussion as to the reason we focused on MtN21. It now reads:

Likewise, only the nodulin MtN21 gene was up-regulated in both lines under weed stress. Thus, it might serve as a tool for identifying signaling networks involved in interference-sensing/response of teosinte, as it likely contains species-specific regulatory elements required for up-regulation in response to weeds. Other coordinately regulated gene sets could also serve to identify such elements but may be cultivar specific.

---

## [Editor Report · Decision Letter 2]

3 Aug 2020

Teosinte (Zea mays ssp parviglumis) growth and transcriptomic response to weed stress identifies similarities and differences between varieties and with modern maize varieties.

PONE-D-20-04184R2

Dear Dr. Horvath,

We’re pleased to inform you that your manuscript has been judged scientifically suitable for publication and will be formally accepted for publication once it meets all outstanding technical requirements.

Kind regards,

Anil Kumar Singh, Ph.D.

Academic Editor

PLOS ONE
---

## [Editor Report · Acceptance letter]

12 Aug 2020

PONE-D-20-04184R2 

Teosinte (*Zea mays ssp parviglumis*) growth and transcriptomic response to weed stress identifies similarities and differences between varieties and with modern maize varieties. 

Dear Dr. Horvath:

I'm pleased to inform you that your manuscript has been deemed suitable for publication in PLOS ONE. Congratulations! Your manuscript is now with our production department. 

Kind regards, 

on behalf of

Dr. Anil Kumar Singh 

Academic Editor

PLOS ONE